# Novel *Stenotrophomonas maltophilia* Bacteriophage as Potential Therapeutic Agent

**DOI:** 10.3390/pharmaceutics14102216

**Published:** 2022-10-18

**Authors:** Rima Fanaei Pirlar, Jeroen Wagemans, Fabian Kunisch, Rob Lavigne, Andrej Trampuz, Mercedes Gonzalez Moreno

**Affiliations:** 1Charité—Universitätsmedizin Berlin, Corporate Member of Freie Universität Berlin and Humboldt-Universität zu Berlin, Center for Musculoskeletal Surgery, Augustenburger Platz 1, 13353 Berlin, Germany; 2Berlin Institute of Health at Charité—Universitätsmedizin Berlin, BIH Center for Regenerative Therapies (BCRT), Charitéplatz 1, 10117 Berlin, Germany; 3Department of Biosystems, KU Leuven, Kasteelpark Arenberg 21, 3001 Leuven, Belgium; 4Faculty of Medicine, Westälische Wilhelms-Universität Münster, Domagkstraße 3, 48149 Münster, Germany

**Keywords:** *Stenotrophomonas maltophilia*, antibiotic resistance, biofilm-associated infection, bacteriophage-antibiotic combination, isothermal microcalorimetry

## Abstract

A novel bacteriophage CUB19 specific to the bacterial species *Stenotrophomonas maltophilia* was isolated from hospital sewage and characterized as a new species belonging to a proposed new phage genus ‘Cubvirus’ (*Caudoviricetes*). Its genome contains a total of 48,301 bp and 79 predicted genes, among which some have been associated with packaging and lysis-associated proteins, structural proteins, or DNA- and metabolism-associated proteins. No lysogeny-associated proteins or known virulence proteins were identified on the phage genome. CUB19 showed stability over a wide range of temperatures (−20 °C–60 °C) and pH values (pH 3–pH 13). Despite its narrow host range, this phage has potent observed antimicrobial and antibiofilm activity. A time-killing curve assay showed significant biofilm reduction after 24 h exposure to CUP19. Isothermal microcalorimetry assays investigating phage-antibiotic combinations revealed the effectiveness of CUB19 during co-administration with increasing antibiotic doses, regardless of the administration approach (simultaneous or staggered). These are encouraging indications for its application as a targeted therapeutic agent against resilient biofilm-associated *Stenotrophomonas* infections.

## 1. Introduction

The prevalence of drug-resistant *Stenotrophomonas maltophilia* is rapidly increasing in nosocomial and community-acquired infections worldwide. It is commonly associated with respiratory infections, but is also found to cause bacteremia, meningitis, endocarditis, pneumonia, osteomyelitis, endophthalmitis, and catheter-related bacteremia/septicemia [1]. Several virulence factors encoded by *S. maltophilia*, including extracellular enzymes (e.g., proteases, esterases, lipases and haemolysin) or cell-associated structures (e.g., lipopolysaccharide and type IV pili), facilitate biofilm formation on abiotic surfaces and host tissues, ultimately leading to the establishment of chronic infections [2]. Due to its ability to adhere to plastic surfaces and form biofilms, it can also colonize many humid hospital surfaces, as well as intravenous cannulas, prosthetic devices, and nebulizers [1]. In fact, *S. maltophilia* has been listed by the World Health Organization (WHO) as one of the leading drug-resistant nosocomial pathogens worldwide, with a mortality rate in immunocompromised patients and severely debilitated individuals that has been increasing in recent years [3,4].

Owing to its low membrane permeability, the presence of chromosomally encoded multidrug resistance efflux pumps, ß-lactamases, and antibiotic-modifying enzymes, *S. maltophilia* strains are intrinsically resistant to a broad range of antibiotics [5], making infections challenging to treat. Indeed, the preferred treatment of *S. maltophilia* infections has been the use of the bacteriostatic compound trimethoprim/sulfamethoxazole (TMP/SMX) or levofloxacin for those intolerant to TMP/SMX [6], but the emergence of resistance to TMP-SMX [5,7] is forcing physicians to consider alternatives. Cefiderocol, a new cephalosporin drug [8], is of great interest in the treatment of MDR Gram-negative infections due to its stability against both serine and metallo-β-lactamases. In *S. maltophilia* isolates, MIC_90s_ values for cefiderocol have been reported as consistently favorable and, moreover, retained susceptibility has been demonstrated in isolates resistant to TMP-SMX, levofloxacin, and polymyxin [9]. However, to date, limited data are available on the antibiofilm activity of cefiderocol in *S. maltophilia* [10].

The use of bacterial viruses, or bacteriophages, as an alternative treatment is an attractive option due to the absence of cross-resistance to antibiotics and mechanisms developed by bacteria to resist antibiotics. Thus, bacteriophages (phages) are considered as an effective solution against MDR (multidrug-resistant), XDR (extensively drug-resistant) and PDR (pan drug-resistant) bacteria [11]. Indeed, phages infect a specific host bacterium by recognition of one or more receptors on the bacterial cell. These receptors can be found in the cell wall, bacterial capsules, slime layers, pili or flagella, often consisting of proteins, lipopolysaccharides, teichoic acids and other cell surface structures serving as irreversible phage-binding receptors [12].

To date, thirty-two phages have been isolated and characterized against *S. maltophilia* presenting a high diversity, nine of which have been experimentally confirmed as virulent phages and hence are desirable for therapeutic use [1]. In this study, we isolated and characterized a novel bacteriophage specific to *S. maltophilia* as a potential therapeutic agent, especially against resilient biofilm-associated infections, and evaluated its combined activity with levofloxacin and cefiderocol.

## 2. Materials and Methods

### 2.1. Bacterial Strains, Antibiotics, and Bacteriophage

This study comprised 40 clinical *S. maltophilia* isolates provided by Labor Berlin—Charité Vivantes GmbH. Cefiderocol powder (Shionogi, Osaka, Japan) and levofloxacin injectable solution (5 mg/mL; Sanofi-Aventis, Frankfurt am Main, Germany) was purchased from the manufacturer. Cefiderocol was reconstituted in sterile 0.9% saline (Merck KGaA, Darmstadt, Germany) right before its use. The medium was supplemented with 25 mg/L glucose-6-phosphate (Merck KGaA, Darmstadt, Germany) for testing of fosfomycin. A novel bacteriophage CUB19 targeting *S. maltophilia* was isolated from hospital sewage from the Charité Campus Virchow-Klinikum and further characterized.

An enrichment method was used for phage isolation as described previously [13]. Four consecutive single-plaque isolation cycles were performed on the bacterial host strain to obtain a single pure phage stock. Thereafter, the phage was propagated in liquid culture, for which *S. maltophilia* bacterial host STM-19 was grown in tryptic soy broth (TSB) (US Biological, Eching, Germany) overnight at 37 °C. Then, one single phage plaque was picked from a plate lysate, resuspended into 1 mL of filter-sterilized SM buffer (10 mM Tris-HCl (Carl Roth GmbH, Karlsruhe, Germany), pH 7.8, 1 mM MgSO_4_ (Merck KGaA, Darmstadt, Germany)), and incubated at 4 °C for 1 h. Subsequently, 0.2 mL of overnight culture was inoculated into 20 mL sterile TSB and incubated with agitation at 37 °C until reaching an OD_600_ of 0.4, after which 0.1 mL SM buffer containing the phage plaque was added. The culture was incubated at 37 °C with agitation for approximately 5 h or until the culture became clear. Finally, the phage lysate was centrifuged at 4000× *g* for 20 min and the supernatant filter-sterilized using a 0.22 μm filter.

Phage precipitation was performed with 8% *w*/*v* polyethylene glycol (PEG-8000; PanReac AppliChem, Darmstadt, Germany) at 4 °C overnight, then centrifuged at 13,000× *g* at 4 °C for 40 min by using a fixed-angle rotor Eppendorf 5810 R centrifuge (Eppendorf, Hamburg, Germany). Afterwards, the pellet was resuspended in SM buffer overnight, filtered and kept at 4 °C until further use. Titration was performed on the host strain to determine the corresponding phage titer.

### 2.2. Morphological Analysis by Transmission Electron Microscopy

The morphology of CUB19 was detected by transmission electron microscopy (TEM) using a Zeiss EM 906 microscope (Carl Zeiss Microscopy Deutschland GmbH, Oberkochen, Germany) and negative staining as in a previous study [14]. Shortly, 15 µL of phage solution was dropped onto Parafilm and transferred onto a carbon-coated and glow discharged (Leica MED 020, Leica Microsystems, Wetzlar, Germany) Ni-mesh grid (G2430N; Plano GmbH, Wetzlar, Germany), and left to adsorb for 10–15 min at room temperature. After washing the grids three times with Aquadest, they were treated with 1% aqueous uranyl acetate (SERVA Electrophoresis GmbH, Heidelberg, Germany) for 20 s for negative staining. Grids were air dried and then imaged by TEM at a voltage of 80 kV. Phage size was measured using the image processing program ImageJ.JS 1.53 m [15].

### 2.3. Genome Extraction, Sequencing, and Annotation

Phage genome isolation was performed as previously described [16]. Briefly, 1 mL of phage stock was treated with 10 µg DNase I and 50 µg RNase A (Roche Diagnostics; Mannheim, Germany) in the presence of MgCl_2_ to degrade the bacterial nucleic acids that are still present after phage lysis, followed by 50 µg/mL of proteinase K (Thermo Scientific, Waltham, MA, USA), 20 mM EDTA and 0.5% SDS treatment to inactivate the DNase I/RNase A and disrupt the viral capsid. Subsequently, extraction by phenol-chloroform (Carl Roth GmbH, Karlsruhe, Germany) was performed to remove debris. The nucleic acid pellet was precipitated (14,000× *g*, 20 min) in the presence of absolute alcohol (Merck KGaA, Darmstadt, Germany) and washed with 70% alcohol before being suspended in deionized distilled water. Nanodrop measurements (Peqlab; Erlangen, Germany) were done to determine concentration and purity (260/230 ratio).

Sequencing was performed on an in-house Illumina (San Diego, CA, USA) MiniSeq instrument. The Nextera Flex DNA library kit (Illumina) was used for the library preparation. Long reads were generated using a Nanopore (Oxford, UK) MinION device equipped with an R.9.4.1 flowcell. The latter library was prepared with the Rapid barcoding. After assembly of the raw sequencing data using Unicycler v3.6.6 [17] on the PATRIC server [18], the most related phages were identified with BLASTn v2.13.0 [19] and Viptree v1.9 [20]. VIRIDIC [21] was used for taxonomic classification. Next, annotation was performed with RASTtk v1.3.0 [22] on the PATRIC server followed by manual curation using BLASTp v2.13.0 (and HMMER v.3.3 [23] or Phyre2 v2.0 [24] in specific cases). A genome map was visualized with Easyfig v2.2.2 [25]. The data were submitted to the NCBI GenBank under accession number OM638088.

### 2.4. Structural Proteome Analysis

Capsid proteins were isolated using a methanol-chloroform extraction starting from 1 mL of the high titer phage stock as previously described [26]. The protein pellet was dissolved in 10 µL SM buffer. Subsequently, 25 µL of freshly prepared denaturation/reduction buffer (6 M urea (Acros Organics, Geel, Belgium), 5 mM dithiothreitol (Cytiva, Marlborough, MA, USA), 50 mM Tris.HCl pH 8 (Sigma-Aldrich, St. Louis, MO, USA) was added and the sample was incubated at 56 °C. After 1 h, the sample was removed from the water bath and allowed to cool down to room temperature. Next, 25 µL of a freshly prepared alkylation buffer (100 mM iodoacetamide (Acros Organics, Geel Belgium), 50 mM NH_4_HCO_3_ (Sigma-Aldrich, St. Louis, MO, USA)) and 150 µL of a 50 mM NH_4_HCO_3_ solution were added to the mixture, which was incubated at room temperature for 45 min in the dark, with brief vortexing steps every 15 min. Finally, after the addition of 0.8 µg trypsin (40 µL of a 20 µg/mL stock solution; Promega, Madison, WI, USA), the proteins were trypsinized overnight at 37 °C. The final peptide sample was stored at −20 °C before analysis by tandem mass spectrometry (MS/MS).

The latter was done in a data-dependent mode on a Q Exactive Plus mass spectrometer (Thermo Fisher Scientific, Waltham, MA, USA) connected to a nanoAcquity UPLC system (Waters Corporation, Zellik, Belgium). Of the peptide mixture, 0.5 µg was loaded on a 200 cm micro pillar array column (mPAC™, PharmaFluidics, Ghent, Belgium) retrofitted to a NanoSpray Flex source. The peptides were eluted at a flow rate of 750 nL/min for 120 min using the following gradient: 1% to 40% acetonitril (ACN) (Thermo Fisher Scientific, Waltham, MA, USA) in 0.1% formic acid (FA) (Acros Organics, Geel, Belgium)/H_2_O for 80 min, 40% to 100% ACN for 5 min, 99% to 1% ACN for 3 min and 34 min at 1% ACN in 0.1% FA/H_2_O. They were then transferred to the gaseous phase with positive ion electrospray ionization at 1.9 kV. Precursors were targeted with an *m*/*z* 0.8 isolation window. A single MS1 scan was performed at a mass resolution of 70,000, an automatic gain control (AGC) target of 3 × 10^6^ ions and a maximum C-trap fill time of 100 ms. The precursor ions were selected “on-the-fly” and fragmented in high-energy collisional dissociation (HCD) mode with a normalized collision energy of 28%. Subsequently, 20 MS/MS scans were performed at a resolution of 17,500, an AGC target of 1 × 10^5^ ions and a maximum injection time of 80 ms. A dynamic exclusion list of 20 s was applied.

The RAW files were exported and processed in MaxQuant version 1.6.17. The files were searched using the target-decoy matching using the proteome of CUB19 as the database, with the false discovery rate set at 1%. Trypsin was indicated as the digestion enzyme and up to two miscleavages were allowed. Carbamidomethylation was set as a fixed modification. Results were uploaded in Perseus 1.6.13.0, and the protein identifications were filtered for reverse sequences, potential contaminants, and those only identified by site. 

### 2.5. Adsorption and One-Step Growth Curve

An adsorption curve assay was performed to determine the adsorption rate of CUB19 as previously described [27]. Briefly, *S. maltophilia* was grown in 10 mL TSB at 37 °C while shaking (150 rpm/min) to an OD_600_ of 0.4 (corresponding to approx. 10^7^ CFU/mL). CUB19 was then mixed with the bacterial strain at MOI = 0.01 (10^5^ PFU/mL) and incubated at 37 °C. Aliquots of 1 mL were taken at 5 min intervals for 35 min and centrifuged at 4000× *g* for 1 min to pellet adsorbed phages. Serial dilutions of the supernatant (non-adsorbed phage) were titrated to determine the number of non-adsorbed phage particles at each timepoint. The adsorption rate was estimated by fitting the logarithmically transformed relative phage concentration data with a linear regression line. The adsorption rate (k) was then calculated by dividing the inverse value of the slope by the concentration (B) of host bacteria (k = −slope/B) [28,29].

To calculate the phage latent period and burst size, a one-step growth curve was performed. Exponentially growing bacteria were infected with CUB19 (MOI of 0.01) and incubated for 20 min at 37 °C to allow phage particles to adsorb. The culture was then centrifuged at 7000× *g* for 10 min and the pellet was resuspended in 10 mL of TSB and kept at 37 °C. Aliquots were removed at 10 min intervals for 150 min and titrated against the host bacterium. The latent period was defined as the time taken for the phage particles to replicate inside the bacterial cells (i.e., from adsorption to first cell burst). The burst size was defined as the number of phage particles released from a single infected bacterial cell and was calculated by dividing the number of phages formed during the rise period with the estimated number of infected cells present in the culture at the latent period time (assuming that infected cells do not multiply or lyse from infection until the rise period), as described by Nabergoj et al. [30].

### 2.6. Host Range and Efficacy of Plating

The phage host range was evaluated against 40 *S. maltophilia* and 43 *Pseudomonas aeruginosa* strains by a soft agar (TSB + 6 g/L agar (Sigma-Aldrich, St. Louis, MO, USA) overlay spot assay. A bacterial lawn was prepared by mixing 50 µL of an overnight grown bacterial culture with 3 mL soft agar, pouring it over tryptic soy agar (TSA) (TSB + 15 g/L agar) and letting it dry for 10 min. Tenfold phage dilutions (10^−1^ to 10^−9^) were then spotted (5 µL) on the bacterial lawn and let to dry. Single phage plaque identification was performed after overnight incubation at 37 °C. Bacterial strains were classified as susceptible to the phage when single phage plaques were visible within any of the dilutions.

The bacterial susceptibility to CUB19 was evaluated by the efficacy of plating (EOP) as previously described [31]. The EOP value was calculated as the ratio between the plaque forming units (PFU) on the tested clinical strains with respect to the isolation/propagation host bacterium (EOP = phage titer on test bacterium/phage titer on host bacterium). EOP values of >0.5 were ranked as ‘high’ efficiency; 0.2–0.5 as ‘medium’ efficiency; 0.001–0.2 as ‘low’ efficiency; 0.0 was considered as ineffective against the target strain [32].

### 2.7. Thermal and pH Stability

The thermal stability of CUB19 was assessed incubating 10^8^ PFU/mL phage solution at different temperatures (−20 °C, 4 °C, 25 °C, 37 °C, 50 °C and 60 °C) for 1 h or 24 h. Likewise, 10^8^ PFU/mL of phage solution was incubated in phosphate-buffered saline adjusted to different pH, ranging from 1 to 13 at room temperature (25 °C), for 1 h and 24 h to evaluate the phage stability at different pH levels. Phage samples were titrated on the host bacterium (STM-19) using the soft agar overlay technique. The experiment was carried out twice as independent biological replicates.

### 2.8. Antimicrobial Susceptibility Test by Isothermal Microcalorimetry

Isothermal microcalorimetry was used to determine the antimicrobial activity of CUB19 at different MOIs (0.1 to 1000) against the host bacterium, as previously reported [33]. Glass ampoules containing 0.8 mL of fresh TSB were inoculated with 0.1 mL of 5 × 10^5^ CFU/mL bacteria and 0.1 mL of phage (at titers ranging from 10^4^ to 10^8^ PFU/mL) and introduced into the calorimeter where heat-flow production was monitored at 37 °C for 48 h. A growth control (GC) without phage and two negative controls containing TSB only (NC) or TSB and phage (PC) without bacteria were included in every test. The experiment was performed in triplicate. Plots of heat-flow (µW) or heat (J) over time (h) of representative samples were prepared using the GraphPad Prism 6 software (GraphPad Software, La Jolla, CA, USA). The lag phase (λ, h) was determined as previously reported [34]. λ refers to the duration of the lag phase measured as the time from the start of the experiment to the interception of a line tangent to the maximum growth rate point and the baseline.

### 2.9. Biofilm Time-Killing Assay

Bacterial biofilms were formed on sterile 4 mm sintered porous glass beads (ROBU, Hattert, Germany) by placing each bead in a single well of a 24-well plate (Corning Inc., Corning, NY, USA) containing 1 mL TSB inoculated with 1:100 dilution of *S. maltophilia* and incubated at 37 °C and 150 rpm for 24 h. Glass beads were then washed with sterile 0.9% saline three times to remove non-adhered planktonic cells and introduced into microcentrifuge tubes with 1 mL of fresh TSB broth inoculated with CUB19 (10^8^ PFU/mL). Tubes were incubated at 37 °C for 0 h, 2 h, 4 h, 6 h, 8 h and 24 h and then biofilm-embedded cells were recovered by sonication of the glass beads in an ultrasound bath at 40 kHz and 0.2 W/cm^2^ (BactoSonic, BANDELIN electronic, Berlin, Germany) for 10 min. Ten-fold serial dilutions of the sonicated suspension were prepared and plated on TSA. After 18 to 24 h incubation at 37 °C, the recovered bacterial cell colonies were counted. Two biological replicates with technical duplicates were carried out. Data was expressed as mean ± standard deviation (SD) and plotted as bacterial count (CFU/mL) over time (h) using the GraphPad Prism 6 software.

### 2.10. Phage-Antibiotic Combinations against Biofilm

First, the minimal inhibitory concentration (MIC) and minimal bactericidal concentration (MBC) for levofloxacin and cefiderocol were determined by the micro-broth dilution assay [35] in cation-adjusted Müller-Hinton broth (Sigma-Aldrich, St. Louis, MO, USA). Susceptibility interpretation was determined according to the Clinical and Laboratory Standards Institute (CLSI) [35]. Consequently, *S. maltophilia* was considered susceptible to levofloxacin when MIC ≤ 2 µg/mL and to cefiderocol when MIC ≤ 4 µg/mL.

Then, 24 h-old-biofilms were formed on porous glass beads as previously described, washed in 1 mL 0.9% saline, and exposed to either phage (10^8^ PFU/mL) or antibiotic at 10× MIC and 100× MIC concentrations prepared in fresh MHB in a final volume of 1 mL and incubated at 37 °C for 24 h. For phage/antibiotic combinations, biofilms were exposed to either a simultaneous co-incubation with phage and antibiotic for 24 h or to a staggered administration adding first phage for 6 h at 37 °C after which the antibiotic was added and incubated for further 18 h at 37 °C. After an overall incubation of 24 h, treated biofilm-beads were rinsed three times with 0.9% saline, placed in sterile glass ampules with 1 mL fresh TSB, and inserted into the calorimeter, where heat produced by viable bacteria present in the bead after 24 h of treatment or no treatment (growth control) were monitored for 48 h. Experiments were performed at least as four biological replicates. Plots of heat flow (µW) over time (h) of representative samples were prepared using the GraphPad Prism 6 software (GraphPad Software, La Jolla, CA, USA).

## 3. Results

### 3.1. Bacteriophage Characterization

The morphology of CUB19 was examined by TEM. The bacteriophage virion consists of an eicosahedral head (101 nm in diameter) and a non-contractile tail (253 nm in length, 16 nm in width) with no visible tail fibers (Figure 1A), corresponding to a siphovirus morphology.

Approximately 95% of CUB19 adsorbed to the host bacterium within 20 min (Figure 1B) with an estimated absorption rate of 1.59 × 10^−9^ phage^−1^ cell^−1^ mL^−1^ min^−1^. The average burst size under standard conditions was estimated to be 155.2 PFU per cell (Figure 1C), calculated dividing the number of phages formed during the rise period (1.55 × 10^9^ PFU/mL) with the estimated number of infected cells (1 × 10^7^ PFU/mL) present in the culture at the latent period time.

The genome of CUB19 was determined using whole genome sequencing. This revealed a dsDNA genome of 48,301 bp, with a GC content of 52.41%. BLASTn only identified two siphoviruses from metagenome data with limited similarity: siphoviruses 94 (Genbank accession number MN855797; 25% query cover; 89% sequence identity) and NHS-Seq1 (MH029512; 40% query cover; 94% identity). An analysis with Viptree indicated that the CUB19 proteome is also related to siphoviruses PMBT14 (NC_048687), Psymv2 (NC_023734), Dina (NC_055026), Bcep176 (NC_007497) and phiE125 (NC_003309). These data confirm the siphovirus morphology as observed by TEM. To classify this phage taxonomically, the intergenomic distance to all related bacteriophages was calculated and plotted (Appendix A). *Stenotrophomonas* phage CUB19 can be defined as a novel phage species belonging to the proposed phage genus ‘Cubvirus’ of yet unclassified family within the *Caudoviricetes*.

To determine the phage genome ends, the genome coverage of the raw reads was inspected in more detail according to Merrill et al. [36]. However, no clear drops or increases in coverage could be identified since the Illumina Nextera flex kit was used for sequencing. Also, the terminase protein did not clarify the packaging strategy of CUB19, since it was not very similar to other terminases with known packaging strategies [36]. Next, long sequencing reads were generated and analyzed, indicating that CUB19 uses a headful packaging strategy. Consequently, the terminase gene was arbitrarily chosen as the genome start.

Structural annotation of the genome identified 79 coding sequences and six tRNAs. Twenty-five coding sequences could be assigned a putative function, leaving 68.4% as dark matter (Figure 2). No lysogeny-associated proteins were identified, which was confirmed by the Phage.AI lifecycle predictor. Moreover, no known virulence proteins were encoded on the phage genome, making it potentially suitable for phage therapy use.

Next, mass spectrometry was performed to determine the virion-associated proteins (Table 1). In total, sixteen CUB19 proteins were detected, of which five were previously assigned a hypothetical function. The endolysin, gp68, was the only protein identified to be involved in cell lysis. Only gp14 and gp23, which were predicted to be the tail assembly chaperone and the putative tail tip assembly protein I, respectively, could not be detected by mass spectrometry, as they are presumably not part of the mature viral particle. After identification of the structural proteome, 62% of the total proteome still remains of unknown function.

### 3.2. Stability of CUB19 at Different Temperatures and pH

The thermal and pH stability of phage CUB19 are depicted in Figure 3. Incubation at temperatures ranging from −20 °C to 60 °C, for 1 h or 24 h, revealed no significant reduction in phage titer. A wide pH range, from 1 to 13, was considered for the evaluation of CUB19 pH stability. The results indicated a stable phage titer at pH levels ranging from 3 to 13 up to 24 h. At pH 1 and 2, phages were completely inactivated after 24 h incubation, whereas approx. 5 × 10^6^ to 10^7^ PFU/mL phages were still active after 1 h exposure to those same low pH levels.

### 3.3. Antimicrobial Activity

The CUB19 host range was determined on 40 *S. maltophilia* strains by soft agar overlay spot assay of tenfold serial dilutions. Four strains (STM-1, STM-19 [host], STM-33 and STM-38) were susceptible to CUB19 infection, showing EOP ratios ranging from 0.25 to 1.45 (Table 2), indicative of a medium to high lytic activity. Extended host range analysis using a panel of 43 *P. aeruginosa* strains did not yield successful infections.

The antimicrobial effect of CUB19 at increasing titers against the host bacterium was assessed by monitoring the heat production during 48 h of treated samples in comparison to the untreated growth control (Figure 4). Complete growth inhibition (absence of heat production) was observed at MOI 100 and 1000, whereas at lower MOIs a remarkable delay in the heat-flow (µW) and heat (J) production compared to the growth control could be observed, with longer lag times (Appendix A). The growth control presented a lag time of 14 h, whereas treated samples showed extended lag times of 27 h (MOI 0.1) and 28 h (MOI 1 and MOI 10).

According to the time-killing assay (Figure 5), the highest reduction in bacteria (an approximately 2-log reduction) was observed after 8 h exposure of STM-19 biofilm to CUB19, while a significant reduction was achieved after 6 h exposure. At 24 h after phage addition, however, an increase in CFU count was seen compared to the count at 8 h.

### 3.4. Phage-Antibiotic Combinations against Biofilm

The MIC and MBC values for levofloxacin and cefiderocol (Table 3) were determined prior to the evaluation of phage-antibiotic combinations against biofilm.

Based on the obtained MIC values, all tested strains were considered susceptible to levofloxacin and cefiderocol according to CLSI.

The antimicrobial effect of the phage and antibiotic was assessed as monotherapy or as a combination treatment following two different approaches: (a) simultaneous exposure of biofilm to 10^8^ PFU/mL CUB19 and different concentrations (MIC, 10× MIC and 100× MIC) of antibiotic for 24 h; and (b) staggered exposure of biofilm to 10^8^ PFU/mL CUB19 for 6 h followed by an 18 h-exposure to antibiotic at 1×, 10× and 100× MIC. The different treatment effect for all three approaches (monotherapy, simultaneous and staggered administration) was evaluated by monitoring for the heat produced by biofilm bacteria still viable on the beads (after treatment) re-inoculated in fresh TSB medium for 48 h. Calorimetric curves are displayed in Figure 6 and Figure 7.

Treatment with the individual phage did not provide a significant antimicrobial effect against STM-1 and STM-19 biofilms, as the heat flow production was comparable to untreated samples (growth controls), whereas an earlier onset of heat production could be observed when tested against STM-33 and STM-38. Antibiotic treatment alone generally revealed a dose-dependent activity, where increasing antibiotic concentrations corresponded with increased antimicrobial activity reflected as a delay in heat production compared to the untreated sample. However, no complete eradication of the biofilm could be achieved with either of the two tested antibiotics, even at the highest tested concentration (100× MIC).

The combination of phage with low doses of levofloxacin (1× MIC) revealed an inferior antimicrobial activity compared to the antibiotic alone, while intermediate doses (10× MIC) showed no or minimal improvement in the antimicrobial effect, and the higher antibiotic doses (100× MIC) had no substantial effect against STM-1 but achieved markedly delayed heat production against STM-19 or even complete biofilm eradication against the STM-33 and STM-38 strains.

When combining the phage with cefiderocol, no relevant divergences were observed between the effects observed for the antibiotic alone or in combination with the phage against STM-1, whereas a substantial improvement in antimicrobial activity was detected when the phage was combined with antibiotic at all tested concentrations against STM-19. As for strain STM-33, a longer suppression of the heat production was found when the phage was combined with 100× MIC compared to the antibiotic alone, while combination with 1× MIC revealed a worse effect, which was also the case for STM-38.

Overall, no major differences in the results were obtained between the two approaches, whether simultaneous or staggered exposure was adopted, for both antibiotics.

## 4. Discussion

The isolation and characterization of phages for the treatment of infections caused by the multidrug resistant pathogen *S. maltophilia* opens the door to new therapeutic opportunities, as nosocomial and community-acquired infections are rapidly increasing in prevalence.

Phages are highly specific to their host, not only at a species level but often even at a strain-specific one. Due to the high genetic and phenotypic heterogeneity within *S. maltophilia* populations, as shown in a recent study analyzing an international collection of 1305 isolates from 22 countries [3], and the narrow-spectrum of activity of many phages, the search for novel phages with therapeutic potential appears to be a crucial aspect in the pursuit of more effective treatments. In this study we isolated a novel phage specific to *S. maltophilia* and performed a genetic and phenotypic characterization to evaluate its potential as a therapeutic agent, also in combination with antibiotics against biofilms.

Genome sequencing of phage CUB19 revealed limited similarity (25–40% query cover) with two siphoviruses in the NCBI database, leading to its definition as a novel phage species, belonging to the proposed phage genus ‘Cubvirus’ within the *Caudoviricetes*. No lysogeny-associated proteins or known virulence proteins could be identified on the phage genome, making it potentially suitable for phage therapy application. In addition, structural proteome analysis identified the role of five viron-associated proteins, which were previously assigned as hypothetical. However, more than half of the virus proteome remains of unknown function. To gather further knowledge about this new phage, we carried out a phenotypic characterization in view of exploring the therapeutic potential of CUB19.

This phage presented a narrow host range, infecting four out of the 40 *S. maltophilia* strains tested and none of the *P. aeruginosa* strains, and a productive cycle of approximately 155 phages released per cell after 90 min. Besides, on two strains (STM-33 and STM-38) other than the host strain (STM-19), the phage produced lysates with higher efficiencies of plating, indicating that CUB19 successfully replicates in these hosts.

CUB19 is stable to temperature and has a broad pH tolerance. Considering that a number of factors are known to impact phage structure and infectivity for the formulation of a drug product, among which temperature and pH are critical [37], having a stable phage can facilitate its production and processing. Isothermal microcalorimetry measurements provided real-time data on the interaction between increasing titers of CUB19 and the host strain, revealing a remarkable antibacterial effect even at the lowest MOI tested, with a prolonged lag time, indicative of a decrease in the number of viable bacteria compared to the untreated sample. At a MOI of 100, no bacterial heat production could be detected during the monitoring period, suggesting the eradication of bacterial cells present on the vial. A comparable antibacterial effect was seen by a lytic phage targeting *Escherichia coli* in a similar study [27].

Several genes encoding for virulence factors, including biofilm formation, allow *S. maltophilia* isolates to adhere to surfaces and colonize medical devices and patients [38]. The behavior and therapeutic potential of *S. maltophilia* phages against biofilms are largely unknown. In our study, although we could not annotate any biofilm-degrading enzymes, CUB19 treatment of pre-formed biofilms displayed a reduction level of *S. maltophilia* biofilm (99% reduction) after 8 h exposure, comparable to the levels seen by Pybus CA et al. in their study with tobramycin (99% reduction) and cefiderocol (97% reduction) and superior to the levels seen with the other tested antibiotics (71% to 87% reduction) [10]. However, after 24 h exposure to CUB19, there was a slight increase in CFU counts, possibly indicative of the emergence of bacterial resistance to the phage, enabling repopulation of the biofilm. Furthermore, when strains STM-33 and STM-38 where exposed to CUB19 for 24 h, an early heat flow production compared to the untreated samples could be seen, which might reflect a phage-lysis-mediated biofilm induction, as seen for example in *Shewanella oneidensis* and prophages [39]. Combination therapy often proves to be more advantageous than monotherapy.

The combination of phages and some antibiotics have been shown to be more effective compared to their independent application [40] and even reduce the capacity of bacteria to develop phage and/or antibiotic resistance [41]. However, phage-antibiotic antagonism may also occur, which may be attributable to the antibiotic mode of action, the concentration and/or the time of antibiotic application, among other factors [42]. Using isothermal microcalorimetry we systematically evaluated the antibiofilm activity of phages combined with two different classes of antibiotics against four *S. maltophilia* isolates. Based on these results, combining phage CUB19 with higher rather than lower antibiotic doses generally led to an improved antimicrobial effect, regardless of the administration approach (simultaneous or staggered). The lack of an improved effect in the case of STM-1 may be due to a reduced phage infectivity, as revealed by a lower EOP.

Nevertheless, further pre-clinical and clinical studies are essential to support the development of phage/antibiotic combination therapy. In this study, we provided the first original in vitro data on a new phage species and its potential to target difficult to treat biofilms.

## Figures and Tables

**Figure 1 pharmaceutics-14-02216-f001:**
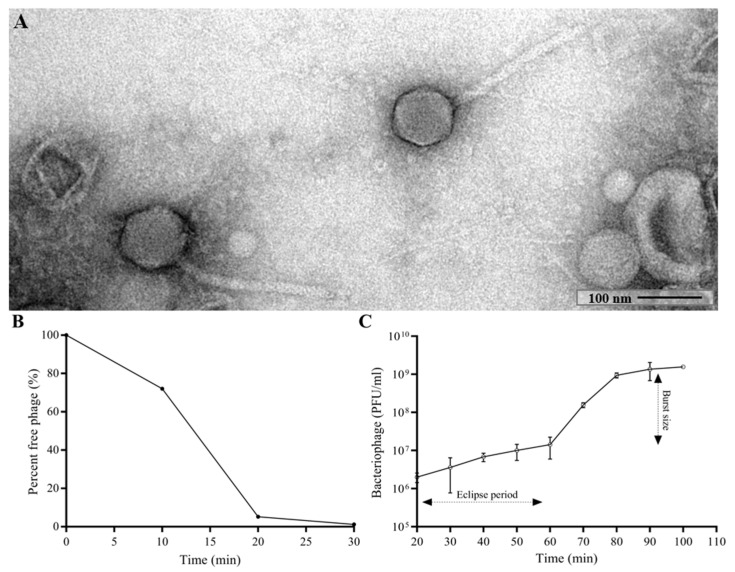
TEM image of the phage CUB19 virion (**A**). Adsorption curve (**B**) and one-step growth curve (**C**) of CUB19 assessed on *S. maltophilia* STM-19 strain. Data are expressed as mean ± standard deviation.

**Figure 2 pharmaceutics-14-02216-f002:**
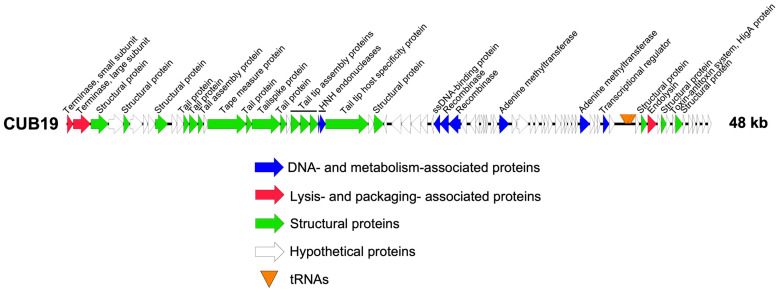
CUB19 genome map. Each arrow represents a coding sequence. In red, genes encoding packaging and lysis-associated proteins are displayed, in green are structural proteins, in blue are DNA- and metabolism-associated proteins and in white are the unknown proteins (adapted from EasyFig).

**Figure 3 pharmaceutics-14-02216-f003:**
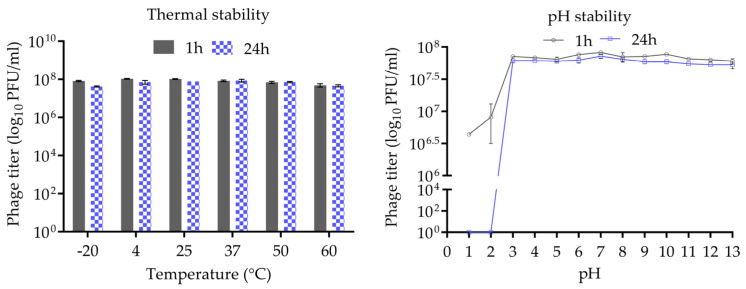
Thermal and pH stability test of CUB19. Error bars represent standard deviations.

**Figure 4 pharmaceutics-14-02216-f004:**
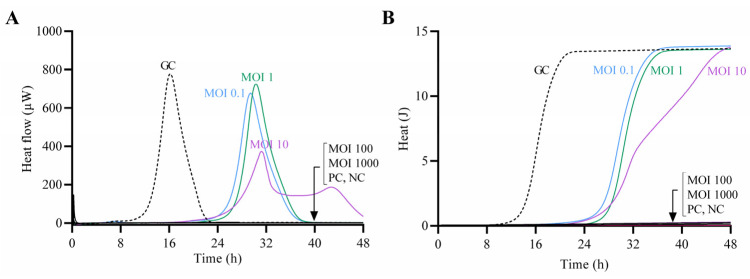
Calorimetric curves. Monitored heat-flow (**A**) and heat (**B**) production of planktonic *S. maltophilia* STM-19 (10^5^ CFU/mL) co-incubated with phage CUB19 at different MOI. GC, growth control; PC, phage control; NC, negative control. Data of a representative experiment are reported.

**Figure 5 pharmaceutics-14-02216-f005:**
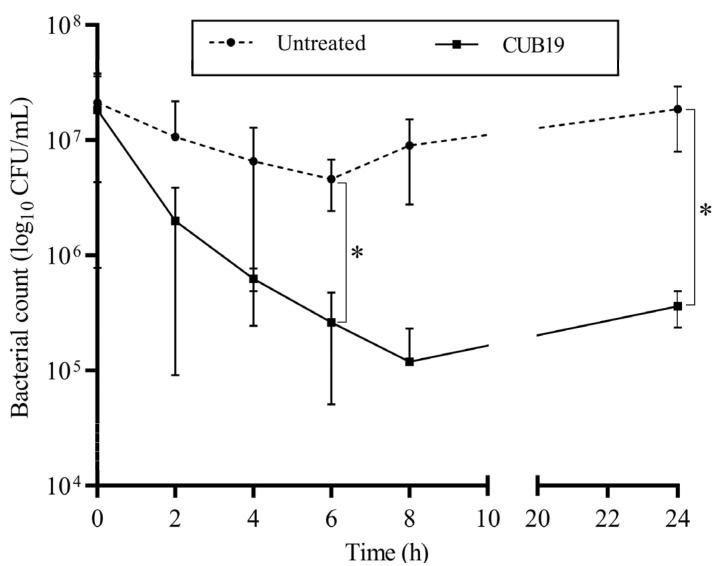
Time-kill curve. *S. maltophilia* biofilm treated with CUB19 (10^8^ PFU/mL) and untreated monitored at 2 h intervals for the first 8 h and after 24 h. Data are expressed as mean ± standard deviation. Results were statistically analyzed using an unpaired t test with Welch’s correction analysis integrated in GraphPad Prism 6; *p*-values <0.05 were considered significant (*).

**Figure 6 pharmaceutics-14-02216-f006:**
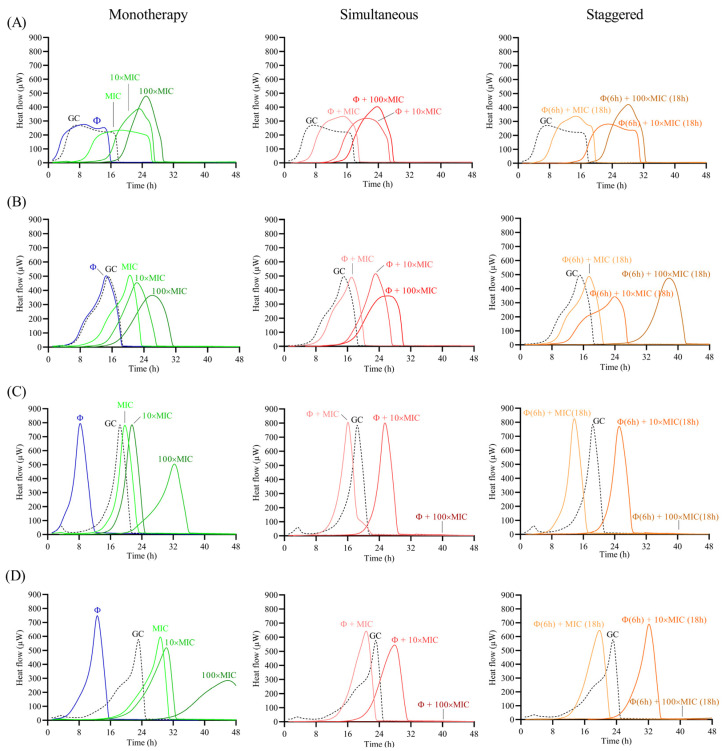
Calorimetric curves. Monitored heat flow (µW) of *S. maltophilia* (**A**) STM-1, (**B**) STM-19, (**C**) STM-33 and (**D**) STM-38 biofilms treated with CUB19 (Φ at 10^8^ PFU/mL) or levofloxacin at different concentrations (1×, 10× or 100× MIC) as monotherapy (left column), or by simultaneous (middle column) and staggered (right column) phage-antibiotic combinations (Φ + levofloxacin). Each curve shows the heat flow produced over time by viable bacteria in the biofilm after 24 h of treatment. GC represents the growth control sample not exposed to any antimicrobials. Data of a representative experiment are reported.

**Figure 7 pharmaceutics-14-02216-f007:**
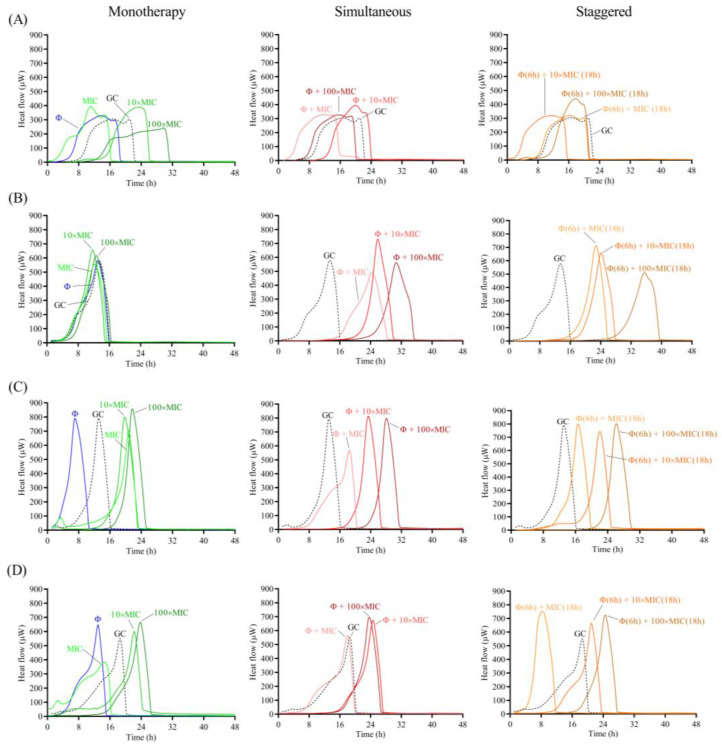
Calorimetric curves. Monitored heat flow (µW) of *S. maltophilia* (**A**) STM-1, (**B**) STM-19, (**C**) STM-33 and (**D**) STM-38 biofilms treated with CUB19 (Φ at 10^8^ PFU/mL) or cefiderocol at different concentrations (1×, 10× or 100× MIC) as monotherapy (left column), or by simultaneous (middle column) and staggered (right column) phage-antibiotic combinations (Φ + cefiderocol). Each curve shows the heat flow produced over time by viable bacteria in the biofilm after 24 h of treatment. GC represents the growth control sample not exposed to any antimicrobials. Data of a representative experiment are reported.

**Table 1 pharmaceutics-14-02216-t001:** Structural proteins of CUB19. Sixteen proteins were identified by ESI-LC-MS/MS as being part of the CUB19 virion. For every protein, the molecular weight (MW), the unique peptide count and the protein sequence coverage is provided.

Protein	Putative Function	MW (kDa)	Unique Peptide Count	Sequence Coverage (%)
gp3	putative portal protein	47.92	7	20.0
gp5	hypothetical protein	15.37	5	43.4
gp9	putative capsid assembly protein F	35.35	3	11.1
gp12	putative tail protein	15.24	1	8.7
gp13	putative tail protein	22.65	5	34.9
gp16	putative tail length tape measure protein H	103.47	19	27.0
gp17	putative tail protein	15.35	1	5.0
gp18	putative tail spike protein	17.95	6	12.7
gp19	putative tail protein	16.51	2	22.3
gp21	putative tail tip assembly protein L	25.72	2	13.8
gp22	putative tail tip assembly protein K	25.10	1	6.8
gp26	putative tail tip host specificity protein J	116.76	16	21.3
gp28	hypothetical protein	27.65	3	17.0
gp67	hypothetical protein	16.49	1	6.2
gp70	hypothetical protein	14.24	3	24.6
gp73	hypothetical protein	19.65	3	34.2

**Table 2 pharmaceutics-14-02216-t002:** Efficiency of plating (EOP) of CUB19 phage dilutions against target bacteria. EOP values were determined using STM-19 as reference strain. Values are expressed as average ± standard deviation.

Strain	EOP	Rank
STM-19	1	reference strain
STM-1	0.25 ± 0.08	medium
STM-33	1.11 ± 0.01	high
STM-38	1.45 ± 0.73	high

EOP values of >0.5 ranked as ‘high’ efficiency; 0.2–0.5 as ‘medium’ efficiency; 0.001–0.2 as ‘low’ efficiency; 0.0 was considered as not effective against the target strain.

**Table 3 pharmaceutics-14-02216-t003:** MIC and MBC values. Determined in planktonic *S. maltophilia* strains by broth microdilution.

Strain	Levofloxacin		Cefiderocol	
	MIC	MBC	MIC	MBC
STM-1	0.5 µg/mL	1 µg/mL	2 µg/mL	4 µg/mL
STM-19	0.5 µg/mL	1 µg/mL	2 µg/mL	4 µg/mL
STM-33	0.5 µg/mL	1 µg/mL	2 µg/mL	4 µg/mL
STM-38	0.5 µg/mL	1 µg/mL	2 µg/mL	8 µg/mL

MIC, minimum inhibitory concentration; MBC, minimum bactericidal concentration.

## Data Availability

Bacteriophage’s whole genome sequence data presented in this study are openly available in the NCBI GenBank under accession number OM638088.

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
