# Peer review of "Novel Stenotrophomonas maltophilia Bacteriophage as Potential Therapeutic Agent"

_pharmaceutics, 2022, doi:10.3390/pharmaceutics14102216_

Round 1

Reviewer 1 Report

Opinion on Pirlar et al.: "Novel Stenotrophomonas maltophilia Bacteriophage as Potential Therapeutic Agent"

The authors describe a new bacteriophage called CUB19 specific to the bacterial species Stenotrophomonas maltophilia. Due to the low level of identity with known phages, a novel genus for the phage is defined, the 'Cubvirus' genus. Host range, temperature and pH stability, burst size and absorption rate is measured for the phage. 16 virion proteins are successfully detected using MS. CUB19 is tested as an anti-biofilm agent alone, or in combinations with varying concentrations of two antibiotics (levofloxacin and cefiderocol), against four strains of S. maltophilia. Phages combined with either antibiotic administered at 100xMIC led to the complete eradication of the biofilm for two bacterial strains, STM-33 and STM-38, irrespective of whether using simultaneously or in a staggered way. This may be useful when setting up future combined therapeutic strategies against S. maltophilia.

The manuscript is well written, the text is easy to read and comprehend. Only a very few typographic errors were found (see below). The results are clearly explained, with one exception: the authors measure the burst size to be 8.2 pfu/cell. However, on Fig. 1C, a much larger burst size is indicated with a double arrow, corresponding to cca. 1E2 pfu/cell. Which value is more relevant or reliable? If it's the 8.2 value, how exactly was it calculated?

Minor errors:

Line 250: 1 ml an -> 1 ml and

L. 257: were monitor -> were monitored

Author Response

Many thanks for the revision and the remarks. The authors have changed the typing errors in the revised manuscript.

Regarding Figure 1C, the authors have indeed noticed a calculation error in the graph for the one step growth curve. We have corrected the error and recalculated both the eclipse period and the burst size of the phage with the new calculations.   

We have therefore also updated plot C in Figure 1.

The authors have also included the information for the calculation of the burst side in the text.

Lines 273 to 276 - The average burst size under standard conditions was estimated to be 155.2 PFU per cell (Figure 1C), calculated dividing the number of phages formed during the rise pe-riod (1.55×109 PFU/ml) with the estimated number of infected cells (1×107 PFU/ml) present in the culture at the latent period time.

Reviewer 2 Report

The manuscript is of interest as it describes in detail a novel bacteriophage CUB19, as a potential therapeutic agent, especially against resilient biofilm-associated infections, and evaluated its combined activity with levofloxacin and cefiderocol. Although this is an interesting study, it is not innovative.

Four strains (STM-1, STM-19 [host], STM-33 and STM-38) were susceptible to CUB19 infection, showing EOP ratios ranging from 0.25 to 1.45 (Table 2), indicative of a medium to high lytic activity. But only three strains in Table 2, why?

Author Response

We thank Referee 2 for the careful revision and for the comment on Table 2. This table refers to the calculated EOP values rather than to the host range. The EOP values are analyzed on the three susceptible strains (STM-1, STM-33, and STM-38) to the phage, using the host strain (STM-19) as reference. We have nevertheless now included STM-19 in the table with an EOP = 1.

Reviewer 3 Report

In this study, a proposed new S. maltophilia bacteriaphage CUB19 was isolated, charactierized and provide an phage-antibiotic combinations against biofilm. Although this manuscript is worth to the field of therapeutic application, the following questions should be clarified first.

1. Antimicrobial susceptibility test, biofilm time-killing assay and phage-antibiotic combinations against biofilm are the main parts to determine the activity of bacteriophage CUB19. So these results should be described in the abstract.

2. The receptors is important in recognition of phages and its host bacterium, which were described in the Introduction. But in this manuscript, the lysis-associated proteins for the bacteria lysis should be described in the Introduction. And the information of lysis-associated proteins should be analyzed and showed in the results.

3. Line 111, please show the enzyme activity of DNase I and RNase A.

4. What is the meaning of the infection cycle I and infection cycle II? And how to determine in Fig. 1C? 

5. The genome map was showed in Fig.2. But in this figure, some important messages were missing, such as the genome scale. I guess the white arrows were for the unknown/other proteins. Please mark and explain in this figure.

6. How to determine the accuracy of the ESI-LC-MS/MS to predict the function of strucural proteins? Take gp16 as an example, please clarify the steps of putative function analysis with only 5% of the sequence coverage.

7. Figure 4 showed an increase in CFU count at 8h after phage addition. Is this mean the development drug-resistance? Please clarify the measures to avoid this influences on the phage-antibiotic combination against biofilm. 

8. P. aeruginosa strains were used to test the extended host range analysis. Did you test other bacteria, such as E.coli, or G+ bacteria?

Reviewer 4 Report

I cannot review this paper unless the phage sequence is provided. The authors appear to have uploaded to NCBI, however the sequence is not live yet, a .gbk file should be sent as Supplementary to the reviewers. I have conveyed these concerns to the handling editor, however no effort was made to correct this.

Author Response

The authors apologize for this issue. As soon as we were informed by the assistant editor of the absence of the phage sequence, we proceeded to provide it.

Round 2

Reviewer 3 Report

   我已经检查了作者的回复。我想这些都能解决我的疑惑。在我看来,这份手稿是值得接受的。

Author Response

The Authors thank Referee 3 for the positive feedback after the manuscript's corrections.

Reviewer 4 Report

Thank you to the authors for making sure the genome sequence was available on NCBI in time for this revision. I will focus my review solely on the genomic aspects in the interest of timeliness and as I believe the other reviewers have done an adequate job commenting on the remainder of the paper.

While the annotation and naming of predicted CDS has been done rather well, I have major concerns regarding the gene-calling performed on the phage nucleotide sequence. The authors have identified 65 CDS and 6 tRNA’s according to the text and actual GenBank entry, however, the authors have missed many valid genes and made some errors in gene directions/start-end coordinates in others. I see the authors have used a RASTtk pipeline for gene-calling. As far as I know, RASTtk uses Prodigal for gene calling, which should work well, so I’m not sure if the authors have made errors after the fact. On my end, the use of Glimmer and Prodigal (via Prokka) on the CUB19 nucleotide sequence produced much higher quality gene-calling than the authors have displayed in the manuscript, and predicted more genes and my preliminary assessment of many of these “additional” genes present in Glimmer/Prodigal gene-calling on my end, suggest they are indeed real CDS. I suggest the authors revisit gene calling, if they don’t have the ability to perform Glimmer/Prodigal gene-calling locally, Prokka is easily and freely available for use on any Galaxy Server (i.e. usegalaxy.org, usegalaxy.eu, usegalaxy.org.au).

I will point out some more notable errors and discrepancies in the authors version of the phage sequence (nucleotide coordinates), however, the authors really need to re-perform gene-calling as mentioned above and fix many other errors themselves:

6,064->4,742. This gene the authors have proposed completely interrupts the operon of the structural module, and the proposed CDS has a very weak ribosome binding site (RBS) and  shows very little similarity to any other genes called on GenBank (only two other genes with very low coverage). Instead, this DNA sequence should represent two individual CDS in the forward direction (i.e. continuing the operon) which are 4,757->5,707 and 5,756->6,094. Not only do these two proposed new genes have much better RBS, but they show similarity to many other genes on GenBank (>100 hits, with very high coverage).

9,977->10,258. This gene is correctly annotated as a tail assembly chaperone. The start and end coordinates are however incorrect. This gene more correctly starts at 9,872, again, a strong RBS is found here, and it continues the operonic structure from the remainder of the structural module. This start site is also more aligned with other similar genes found on GenBank. Secondly, tail assembly chaperones are slightly more complicated to annotate as they are notorious for containing a ribosomal slippage sequence causing a frameshift during transcription. The result of this frameshift causes two different tail assembly chaperone proteins to be produced, the more frequent being a shorter one (corresponding to 9,872->10,258 coding sequence), and a longer one where a frameshift occurs during transcription causing a combined product from two successive “genes” (9,872->10,584). This is well documented phenomenon in this particular protein and many examples can be found online, but this page is a good example (https://cpt.tamu.edu/training-material/topics/phage-annotation-pipeline/tutorials/annotating-tmp-chaperone-frameshifts/tutorial.html). In this case, I strongly believe the slippage occurs between 10,249->10,254 (GGGAAAC) leading to a -1 framing shift allowing expression all the way to the 10,584 generating a ~240 a.a combined chaperone protein.

I trust the authors will perform gene calling again and have another look at the remainder of the phage sequence. 

Author Response

The Authors thanks Referee 4 for his revision. We have recalled the genes using Prokka according to Reviewer 4's comments.